# Impact of cfDNA Reference Materials on Clinical Performance of Liquid Biopsy NGS Assays

**DOI:** 10.3390/cancers15205024

**Published:** 2023-10-17

**Authors:** Ariane Hallermayr, Thomas Keßler, Moritz Fujera, Ben Liesfeld, Samuel Bernstein, Simon von Ameln, Denny Schanze, Verena Steinke-Lange, Julia M. A. Pickl, Teresa M. Neuhann, Elke Holinski-Feder

**Affiliations:** 1MGZ—Medizinisch Genetisches Zentrum, 80335 Munich, Germany; thomas.kessler@mgz-muenchen.de (T.K.); moritz.fujera@mgz-muenchen.de (M.F.); verena.steinke-lange@mgz-muenchen.de (V.S.-L.); teresa.neuhann@mgz-muenchen.de (T.M.N.); elke.holinski-feder@mgz-muenchen.de (E.H.-F.); 2Medizinische Klinik und Poliklinik IV, Campus Innenstadt, Klinikum der Universität München, 80336 Munich, Germany; 3European Liquid Biopsy Society, 20246 Hamburg, Germany; 4Limbus Medical Technologies GmbH, 18055 Rostock, Germany; ben.liesfeld@limbus-medtec.com (B.L.); samuel.bernstein@limbus-medtec.com (S.B.); 5Immune-Oncological Centre Cologne (IOZK), 50674 Cologne, Germany; vonameln@iozk.de; 6Institute of Human Genetics, University Hospital Magdeburg, Otto-von-Guericke University, 39120 Magdeburg, Germany; denny.schanze@med.ovgu.de

**Keywords:** ctDNA, duplex sequencing, cfDNA reference materials, validation, cancer, mosaic disease, liquid biopsy

## Abstract

**Simple Summary:**

Liquid biopsy is a promising tool for the detection of low-frequency variants present in cancer and mosaic disease and is more and more adopted in clinical practice. Still, accurate validation and standardization across laboratories is lacking. Here, we show that the reference materials used during validation have a significant impact on a liquid biopsy next-generation-sequencing (NGS) assays analytical performance evaluation. With our findings, we developed a guide for the selection of suitable reference materials to eventually enable comparable results for patients across different laboratories. Highly sensitive and precise liquid biopsy NGS assays are critical to ensure accurate clinical interpretation.

**Abstract:**

Background: Liquid biopsy enables the non-invasive analysis of genetic tumor variants in circulating free DNA (cfDNA) in plasma. Accurate analytical validation of liquid biopsy NGS assays is required to detect variants with low variant allele frequencies (VAFs). Methods: Six types of commercial cfDNA reference materials and 42 patient samples were analyzed using a duplex-sequencing-based liquid biopsy NGS assay. Results: We comprehensively evaluated the similarity of commercial cfDNA reference materials to native cfDNA. We observed significant differences between the reference materials in terms of wet-lab and sequencing quality as well as background noise. No reference material resembled native cfDNA in all performance metrics investigated. Based on our results, we established guidelines for the selection of appropriate reference materials for the different steps in performance evaluation. The use of inappropriate materials and cutoffs could eventually lead to a lower sensitivity for variant detection. Conclusion: Careful consideration of commercial reference materials is required for performance evaluation of liquid biopsy NGS assays. While the similarity to native cfDNA aids in the development of experimental protocols, reference materials with well-defined variants are preferable for determining sensitivity and precision, which are essential for accurate clinical interpretation.

## 1. Introduction

Liquid biopsy is emerging as a promising tool for the non-invasive detection of genetic variants present in only a subset of cells. It enables the analysis of cell-free DNA (cfDNA) from plasma and other body fluids and allows the stratification of patients who will benefit from targeted therapies [1]. In particular, the detection of circulating tumor DNA (ctDNA), which as part of total cfDNA directly correlates with the presence of tumor cells, appears promising for clinical application in cancer patients [2]. Besides companion diagnostics, which has already been included in clinical guidelines at the international and national level [3,4,5,6,7], it has also been described as a powerful tool for the detection of minimal residual disease (MRD) and monitoring the response or resistance to treatment [8,9,10,11,12,13]. Further, liquid biopsy is not limited to cancer patients, but is also promising for the detection of somatic mosaicism leading to monogenic disease [14,15].

To widely introduce liquid biopsy into clinical practice and to offer its benefits to as many patients as possible, standardization and quality assurance of liquid biopsy analysis across diagnostic laboratories is essential [16,17,18]. Highly sensitive and specific methods are required to detect tumor- or mosaic-specific pathogenic variants, which in most cases are present in plasma at very low variant allele frequencies (VAFs) of <1% [2,19,20]. Reliable clinical interpretation of such low VAFs requires thorough analytical validation based on well-defined reference materials. However, to date, commercial cfDNA reference materials are not native cfDNA, but materials artificially produced to achieve cfDNA-like features [21,22] with different quality metrics.

In this study, we analyzed 44 commercial cfDNA reference samples, including six different types of materials, and 42 cfDNA samples from patients with cancer or mosaic disease using a highly sensitive duplex-sequencing approach in three different laboratories. To identify the reference materials most similar to native cfDNA, we comprehensively evaluated the analytical performance of the different cfDNA reference materials and patient samples. We examined metrics of wet-lab and sequencing quality, which are essential for the development of liquid biopsy next-generation-sequencing (NGS) assays, as well as background noise, which is critical for the specific identification of true positive variants. Eventually, we developed a guideline for the selection of appropriate reference materials for the different steps in performance evaluation.

## 2. Materials and Methods

### 2.1. Study Design

Using duplex-sequencing technology for highly sensitive detection of variants with low VAFs, four custom gene panels targeting clinically relevant regions in cancer or mosaic diseases (Appendix A) were used at three different laboratories (target region sizes: Lab 1: 100 or 73.8 kb, Lab 2: 25.2 kb, Lab 3: 16.3 kb). To evaluate analytical performance, a total of 44 cfDNA reference samples (Lab 1: 28, Lab 2: 10, Lab 3: 6) and 42 patient samples (Lab 1: 28, Lab 2: 14, Lab 3: 14) were analyzed. Analytical metrics were normalized to the target size of the respective gene panel (Figure 1).

### 2.2. Reference Materials

Six types of reference materials were obtained from different providers. All three laboratories analyzed cfDNA isolated from version 1 and 2 of Seraseq^®^ ctDNA Complete™ Reference Material or ctDNA mutation mix (LGC SeraCare, Milford, MA, USA) with spike-in variants (single nucleotide variants, SNVs and small insertions and deletions, InDels) with variant allele frequencies (VAFs) ranging from 0% to 5% in a background of NA24385 (SeraCare). Labs 1 and 2 further evaluated SensID cfDNA reference materials with spike-in variants (SNVs and InDels) with VAFs ranging from 0% to 5% in a background of NA24385 (SensID, Rostock, Germany) (SensID). In addition, Lab 1 analyzed the SeraSeq Myeloid ctDNA Mix with spike-in variants (SNVs) with VAFs from 0% to 0.5% (LGC SeraCare) (SeraCare_Myo) and mechanically fragmented genomic DNA (gDNA) from NA24385 (Coriell Institute, Camden, NJ, USA) (Coriell). Lab 2 further analyzed the OncoSpan cfDNA (Horizon Discovery, Waterbeack, UK) (Horizon). Lab 3 analyzed in addition the Twist cfDNA Pan-cancer reference standard with spike-in variants (SNVs and InDels) with VAFs from 0% to 1% (Twist Bioscience, South San Francisco, CA, USA) (Twist) (Appendix A).

### 2.3. Patient Samples

Plasma samples were collected from 42 patients in Streck Cell-Free DNA tubes (Streck, La Vista, NE, USA) or PAXgene Blood ccfDNA tubes (Qiagen, Hilden, Germany) (Appendix A). Usage of tubes specific for cfDNA stabilization ensure avoidance of increased germline DNA (gDNA) background.

### 2.4. cfDNA Isolation

CfDNA from 3 to 7.5 mL plasma was isolated using the QIAamp circulating nucleic acid kit (Qiagen), the Maxwell^®^ RSC ccfDNA LV Plasma Kit (Promega, Madison, WI, USA), or the NEXTprep-Mag™ cfDNA Isolation Kit (PerkinElmer, Waltham, MA, USA) according to the manufacturer’s instructions. All buffer volumes were adjusted to the respective plasma volumes. All membrane washing steps were performed twice. CfDNA concentration and purity was determined using the High Sensitivity NGS Fragment Analysis Kit on the Fragment Analyzer system (Agilent, Santa Clara, CA, USA) or the Cell-free DNA ScreenTape assay on the TapeStation system (Agilent).

### 2.5. Kit Design

xGen Custom Hyb Panels (Integrated DNA Technologies, IDT, Coralville, IA, USA) were created based on three target regions (Appendix A). A fourth target-region set (Appendix A) was created as a SureSelect XT HS2 custom oligo pool (Agilent Technologies, Santa Clara, CA, USA), and as a Twist custom probes panel (Twist Bioscience, South San Francisco, CA, USA).

### 2.6. Library Preparation and Sequencing

Labs 1 and 2 performed library preparation of cfDNA reference materials and native cfDNA samples from patients using the xGen™ cfDNA & FFPE DNA Library Preparation Kit (IDT) as per manufacturers’ instructions. A total of 25 ng of cfDNA reference materials and 5 to 50 ng of native cfDNA from patient samples, depending on the availability of cfDNA, was used as input material for library preparation (Appendix A). Fragmentation of gDNA NA24385 (Coriell) was performed with a pulsed shearing program on the Covaris E220 system (Covaris, Woburn, MA, USA) in 8 microTUBE-50 AFA Fiber Strip V2 (Covaris) with the following settings: 10 repeats of 10 s treatment with a peak incident power of 75 W, 15% duty factor, and 500 cycles per burst at 7 °C. Libraries from cfDNA and gDNA were amplified with nine PCR cycles. Libraries were quantified using a DNA 1000 Kit on the Bioanalyzer system (Agilent). Following library preparation, target enrichment was performed in accordance with the xGen hybrid-capture of DNA libraries protocol (IDT, v4, May 2019) using the xGen Lockdown Probe Pools (Integrated DNA Technologies, as described in Kit design), the xGen Hybridization and Wash Kit (IDT), and the xGen Library Amplification Primer Mix (IDT). Prior to hybridization, one to seven cfDNA samples or seven gDNA samples were pooled equimolarly. Hybridization was performed over 14 to 16 h and final libraries were amplified with 11 PCR cycles. Pre-pools were quantified using a High Sensitivity DNA Kit (Agilent) on the Bioanalyzer system (Agilent).

Lab 3 measured DNA concentration using a QubitTM 4 fluorometer (InvitrogenTM, Carlsbad, CA, USA). Library preparation was performed using the SureSelect XT HS DNA Reagent Kit (Agilent) according to manufacturers’ instructions. A total of 10 to 100 ng of cfDNA was used as input material for library preparation. Libraries were quantified using a D100 Screen Tape on the TapeStation 4200 system (Agilent). Following library preparation, target enrichment was performed in accordance with the SureSelect XT HS2 Post-Capture protocol using the SureSelect XT HS2 custom oligo pool (Agilent Technologies, as described in Kit design). Alternatively, library preparation of cfDNA reference materials was performed using the Twist EF Library Preparation Kit, UDI indices, universal adapter, and universal blockers (Twist Bioscience) according to manufacturers’ instructions. A total of 10 to 100 ng of cfDNA was used as input material for library preparation. Libraries were quantified using a D1000 Screen Tape on the TapeStation 4200 system (Agilent). Following library preparation, target enrichment was performed in accordance with the Twist EF Pre-Capture protocol using the Twist custom probes panel (Twist Bioscience, as described in Kit design).

Sequencing pools were quantified using a High Sensitivity DNA Kit (Agilent) on the Bioanalyzer system (Agilent) or a D100 Screen Tape on the TapeStation 4200 system (Agilent). Paired-end sequencing with 2 × 151 bp reads was performed on a MiSeq, NextSeq 500, or NovaSeq 6000 system (Illumina, San Diego, CA, USA) with an average sequencing depth of 40,000× to 50,000×.

### 2.7. Bioinformatics Analysis

Raw data (FASTQ.GZ format) were uploaded to the VARVIS^®^ platform and aligned against the hg38 (Labs 1 and 2) or the hg19 (Lab 3) reference genome followed by variant calling using the duplex-sequencing bioinformatics pipeline from the varvis^®^ software version 1.23.2 with in silico validated standard settings. Within the liquid biopsy workflow, the duplex consensus was built by extraction and processing of duplex-sequencing barcodes according to Schmitt et al. [23] and the manufacturer’s “analysis guidelines, version 1” [24]. A minimum of two reads were used to construct a strand-specific consensus read. Strand-specific consensus reads were then combined to create a final consensus read.

### 2.8. Statistical Analysis

Differences between cfDNA reference materials and patient samples were determined using a Wilcoxon test. Bonferroni correction was used to adjust *p*-values for multiple testing. All statistical analyses were performed using statistical functions within the R version 4.3.1.

## 3. Results

### 3.1. Performance Evaluation of Liquid Biopsy NGS Assays

Within this study, we comprehensively evaluated cfDNA reference materials and patient samples in three different laboratories using the duplex-sequencing technology. Lab 1 analyzed four types of cfDNA reference materials and patient samples using two gene panels, Lab 2 analyzed three types of reference materials and patient samples using one gene panel, and Lab 3 analyzed two types of reference materials and patient samples with another gene panel (Figure 1).

### 3.2. Quality Metrics of Library Preparation and Sequencing Performance

To evaluate the comparability of cfDNA reference materials and native cfDNA in the wet-lab, we examined metrics critical for assessing the quality of NGS libraries and sequencing data. To this end, we compared library yield as well as sequencing depth, on-target rate, and GC content of strand-aware consensus reads.

By evaluating the library yield of four types of cfDNA reference materials and native cfDNA at Lab 1, we gained an understanding of the efficacy and comparability of the wet-lab methods. Direct comparison of library yields allowed us to determine the extent to which the reference materials reasonably reflect native cfDNA. We found that, compared to patient samples, the SeraCare reference materials achieved significantly higher library yield (*p*-value = 1.2 × 10^−3^), while the SensID reference materials achieved significantly lower library yield (*p*-value = 1.4 × 10^−3^) (Figure 2A).

Examining the sequencing depth of six types of cfDNA reference materials and patient samples, we aimed to evaluate the extent to which the reference materials capture the variability of native cfDNA. Again, we observed significantly higher sequencing depth in SeraCare reference materials (*p*-value = 2.4 × 10^−2^) and significantly lower sequencing depth in SensID reference materials (*p*-value = 2.8 × 10^−2^) compared to patient samples (Figure 2B). We did not detect significant differences in on-target rates between cfDNA reference materials and patient samples, indicating that all tested cfDNA reference materials did not affect target enrichment (Figure 2C). To investigate the consistency of cfDNA reference materials across different genomic regions, we further assessed GC content and observed a significantly increased GC content in SeraCare cfDNA reference materials (*p*-value = 9.2 × 10^−3^) compared to patient samples (Figure 2D).

Analysis of these key wet-lab metrics allowed us to comprehensively assess the comparability of cfDNA reference materials and native cfDNA. While we achieved significantly lower library yield and sequencing depth with the SensID reference materials compared to patient samples, the SeraCare reference materials achieved significantly higher library yield and sequencing depth. The Coriell reference material showed no significant differences compared to patient samples, yet our data suggest lower library yield, sequencing depth, and on-target rate (Figure 2A–C).

Consequently, the Twist, Horizon, and SeraCare_Myo reference materials were the materials that most closely resembled patient samples and therefore the only samples suitable for wet-lab development.

### 3.3. Distribution of cfDNA Fragment Length

Native cfDNA fragment length profiles typically present with a peak at ~167 bp, which corresponds to the length of DNA bound by a nucleosome plus linker DNA (Peneder et al.). To assess how well cfDNA reference materials represent the fragment length distribution of native cfDNA, we compared the insert size distribution of all six types of cfDNA reference materials with patient samples. We found that the Twist reference material was the only reference material tested with an insert size distribution similar to native cfDNA. The SeraCare reference material as well has a slightly similar insert size distribution compared to native cfDNA. The remaining cfDNA reference materials (Horizon, SeraCare_Myo, SensID, Coriell) show a broad distribution of DNA fragment size comparable to mechanical DNA fragmentation (Figure 3).

Overall, this comparison shows that the Twist cfDNA reference material most closely matches the fragment length distribution of native cfDNA.

### 3.4. Variant Detection Rate at Low VAFs

To comprehensively evaluate the analytical performance of liquid biopsy NGS assays, we investigated the presence of variants with low VAFs in cfDNA reference materials. In general, the number of such low VAF variants per kilobase (kb) is examined to determine the background noise. The limit of blank (LOB) is determined based on the background noise and describes the cutoff value used to distinguish true-positive from false-positive variants. Accordingly, it is a critical cutoff for the clinical interpretation of whether an actionable variant has been detected. The limit of detection (LOD) describes the VAF above which variants are detected with a sensitivity of 95% [16,17,25].

By comparing the number of low VAF variants identified per kb of the target region, we found that the different cfDNA reference materials had significantly different numbers of such variants. We determined the number of variants within different VAF intervals, ranging from 0.01% to 0.1%, from 0.1% to 0.25%, from 0.25% to 0.5%, from 0.5% to 1%, and from 1% to 5% VAF. The Twist and SeraCare reference materials showed significantly higher numbers of variants compared to the patient samples in all five VAF intervals (Twist: *p*-values: 2.6 × 10^−3^–4.3 × 10^−3^, SeraCare: *p*-value: 2.4 × 10^−10^–3.3 × 10^−4^). The SensID reference material had a significantly lower number of variants compared to the patient samples in the VAF interval ranging from 0.01% to 0.1% (*p*-value = 8 × 10^−4^). The Horizon, SeraCare_Myo, and Coriell reference materials contained a number of variants comparable to the patient samples in all VAF intervals (Figure 4A).

The significant differences in low VAF variants observed in different cfDNA reference materials and the low number of such variants in patient samples indicate that the variants detected are true positives that are present in the reference materials as a result of the manufacturing process. Comparing the median number of variants per kb of target region above different LOBs, we found that a higher number of variants is observed in the Twist, SeraCare, and Horizon reference materials compared to patients even at a cutoff of 0.25% VAF. For the SeraCare_Myo, SensID, and Coriell reference materials, we found that fewer variants are detected than in patient samples at a cutoff of only 0.01% VAF. A cutoff of 0.1% VAF increases the similarity between these reference materials and patient samples (Figure 4B). Accordingly, we were able to establish the LOB of all our liquid biopsy NGS assays at a VAF of 0.1%.

Overall, the comparison of variant calls between the cfDNA reference materials and the patient samples provided valuable insight into the true clinical utility of our liquid biopsy NGS assay. Our analysis showed concordance of variant detection rates between the SeraCare_Myo, SensID, and Coriell reference materials and the patient samples, confirming the reliability of duplex-sequencing technology for detecting the variants with the lowest VAFs. Discrepancies in the number of variants identified per kb between patient samples and Twist and SeraCare cfDNA reference materials emphasized potential origination of variants during the manufacturing process.

Hence, for validation of sensitivity and precision of a liquid biopsy NGS assay, the SeraCare_Myo, SensID, and Coriell materials were determined as suitable cfDNA reference materials.

### 3.5. Guideline for the Selection of Reference Materials

Our data indicate that different types of reference materials are required for various steps during analytical validation of liquid biopsy NGS assays because no single cfDNA reference material matches native cfDNA in all important parameters. We therefore developed a guide for the selection of appropriate cfDNA reference materials for each step of the performance evaluation. While comparable fragment size distribution, library yield, consensus sequencing depth, on-target rate, and GC content between reference materials and native cfDNA are critical for performance evaluation of library preparation and sequencing quality, a well-defined background and spike-in variants with known VAFs are critical for bioinformatics validation. We demonstrated that the SeraCare_Myo, Twist, and Horizon cfDNA reference materials best represent patient samples with respect to wet-lab and sequencing quality criteria. In addition, the Twist and SeraCare cfDNA reference materials have the most similar fragment size distribution to the patient samples. When determining precision, the SeraCare_Myo, SensID, and Coriell reference materials had a well-defined variant background and thus the highest concordance with the patient samples. In determining sensitivity, the presence of well-defined spike-in variants with known VAFs is critical. Here, the reference materials from SeraCare and SensID appear to be the most promising (Figure 5).

We followed this guideline and, using the SensID wild-type cfDNA reference material, established the LOB of all our liquid biopsy NGS assays at 0.1% VAF as cutoff for specific variant detection. Using the SeraCare and SensID reference materials with spike-in variants with 0.5% VAF, we established the LOD as cutoff for sensitivity of our liquid biopsy NGS assay at 0.5% VAF. Using unsuitable reference materials, especially for determination of the LOB, would eventually lead to a higher cutoff and therefore an overall lower sensitivity of liquid biopsy NGS assays. To achieve the highest possible diagnostic yield, a high sensitivity even at low VAFs is essential.

## 4. Discussion

Liquid biopsy holds great potential in revolutionizing clinical diagnostics and precision medicine. It enables the non-invasive detection of genetic variants present only in a subset of cells, making it a promising tool in diagnosis of cancer and mosaic diseases [3,8,10,13,14,15,26]. Liquid biopsy is currently on the verge of becoming a routine clinical application for companion diagnostics in cancer, as it has been incorporated in clinical guidelines on the international and national level [3,4,5,6,7]. In addition, health insurance companies have started to reimburse the cost of liquid biopsy analyses. However, there are still some challenges to overcome before liquid biopsy-based diagnostics can become part of standard patient care. Apart from the high costs associated with the ultra-deep sequencing required to detect the variants with the lowest VAFs, the use of such high-sensitivity assays requires standardization to ensure high-quality results for patients. To this end, ctDNA analysis has recently been included in the German national guideline for quality assurance in laboratory medicine [27]. The use of reference materials with well-defined background and spike-in variants is crucial for validation of liquid biopsy NGS assays and quality assurance across laboratories. In our study, we critically evaluated the impact of six different types of cfDNA reference materials on the analytical performance evaluation of liquid biopsy NGS assays at three different sites.

A major challenge for standardization of liquid biopsy NGS assays across laboratories is the lack of reference materials resembling native cfDNA. Through a comprehensive evaluation of wet-lab procedures, sequencing quality, and variant detection in cfDNA reference materials and patient samples, we intended to improve our understanding of the applicability of these materials in the context of analytical performance evaluation of liquid biopsy NGS assays. Our results showed varying degrees of comparability between reference materials, indicating the complexity of capturing the biological characteristics of native cfDNA. Using a multidimensional assessment, we found the highest degree of agreement with patient samples in terms of wet-lab procedures and sequencing quality in SeraCare_Myo reference materials. When considering fragment size distribution, the Twist material was the only reference material that we tested that closely matched the profile of native cfDNA. In terms of the number of low VAF variants detected, the SeraCare_Myo, Coriell, and SensID reference materials showed the highest agreement with the patient samples. All reference materials except the Horizon material have a background of NA24385 DNA. However, significant differences in variant calls were observed between these materials, suggesting that differences in manufacturing processes introduce low-frequency variants in some of these materials. For example, for the preparation of the SeraCare reference material, an amplification step is performed to achieve a fragment size distribution similar to that of native cfDNA [21]. It is known that errors in amplification lead to artifacts that present as low-frequency variants in sequencing data. Therefore, when determining the precision of an assay, it is critical to consider those reference materials that do not involve amplification during manufacturing. Although no reference material showed high concordance with native cfDNA for all parameters, different reference materials can be used to evaluate specific analytical targets. The integration of such reference materials into the validation of diagnostic workflows is essential to improve the reliability and clinical utility of liquid biopsy technologies.

Ensuring high specificity and precision of liquid biopsy NGS assays and standardization across laboratories is the basis for implementing such tests into clinical practice. However, analytical validity is not the only consideration that needs to be taken into account, but also the clinical utility of such an assay. In our study, we found that improving the cutoff value for variant detection based on reference materials that match the number of variants detected in patients has the potential to increase the sensitivity of liquid biopsy NGS assays while maintaining high precision. Ensuring highly sensitive and specific detection of pathogenic variants present in low VAFs supports the clinical interpretation of liquid biopsy reports. Therefore, in addition to careful selection of reference materials to be used in analytical validation, regular participation in proficiency testing is essential for quality assurance [16,17,27,28].

While the results of our study are informative regarding the importance of carefully considering the advantages and disadvantages of different commercial reference materials for analytical validation of liquid biopsy NGS assays, our study has some limitations. Library preparation and sequencing were performed at three different labs using four different custom-designed target panels and different laboratory workflows, potentially impacting the comparability of the resulting NGS data. Thus, we normalized the number of variants identified in each sample to the respective target regions. Another limitation is the fact that not all types of reference materials and patient samples were processed at each participating lab. However, similar results obtained for those reference materials that were indeed sequenced at multiple labs support the comparability of data between laboratories. Further, using a wide range of input material from 5 to 50 ng of native cfDNA from patient samples might introduce bias for variant detection. We still decided to use all available cfDNA material from patients to provide the best possible care. Statistically, it is likely that higher amounts of input material might lead to higher detection rates for low-frequency variants. However, since we observed comparable detection rates for low-frequency variants across patient samples and since for all reference materials a standardized amount of 25 ng of input DNA was used, we strongly believe that these differences do not impair the findings of our study.

## 5. Conclusions

In conclusion, our study emphasizes the significant differences between commercial cfDNA reference materials. While such reference materials are critical for analytical performance evaluation of liquid biopsy NGS assays, consideration of the advantages and disadvantages of each material is essential. Establishing robust cutoffs for variant detection is critical for the reliable clinical interpretation of liquid biopsy-analysis results. The use of standardized analytical validation protocols is essential to achieve a comparable diagnostic yield between different labs. With our study, we demonstrate the impact of different reference materials on the analytical performance of an assay and thus contribute to the necessary standardization of liquid biopsy NGS assays for implementation into clinical practice. Increasing evidence on clinical utility of liquid biopsy diagnosis for cancer and mosaic disease raises the importance of such tests and will eventually lead to implementation into clinical guidelines and routine diagnostics.

## Figures and Tables

**Figure 1 cancers-15-05024-f001:**
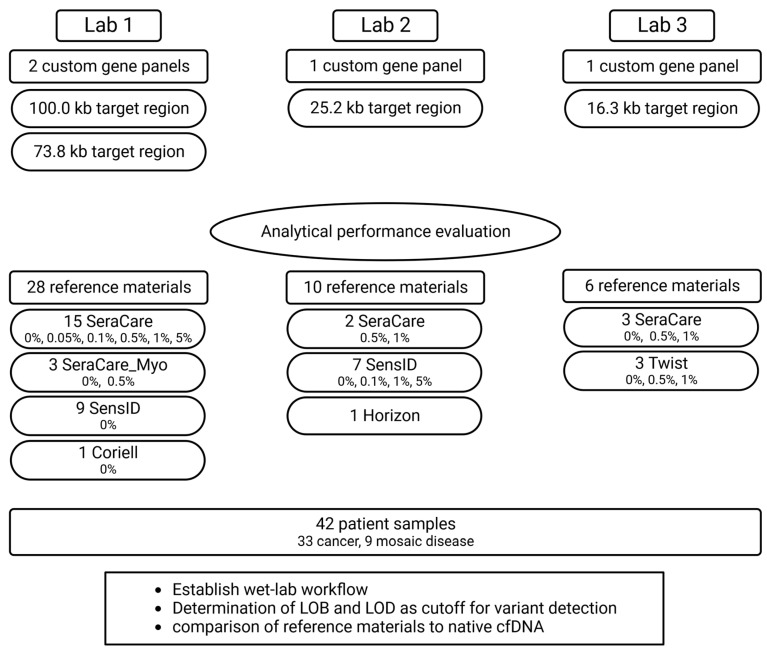
Study outline. The analytical performance of liquid biopsy NGS assays was evaluated at three different labs by analyzing 44 commercial cfDNA reference samples and 42 patient samples. (SeraCare = SeraSeq ctDNA complete reference material, SeraCare_Myo = SeraSeq Myeloid ctDNA mix, SensID = SensID cfDNA reference material, Coriell = NA24385 gDNA provided by Coriell, Horizon = OncoSpan cfDNA, Twist = Twist Pan-cancer reference standard, LOB = limit of blank, LOD = limit of detection).

**Figure 2 cancers-15-05024-f002:**
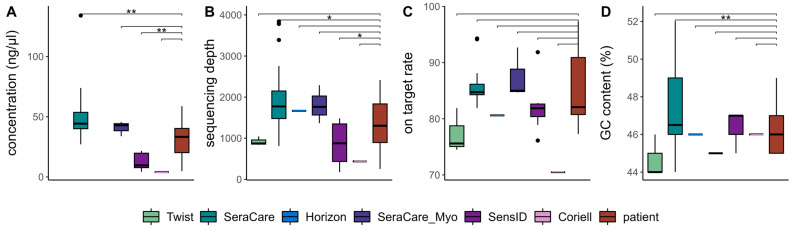
Wet-lab and sequencing quality metrics for six commercial reference materials and patient samples: (**A**) library yield, (**B**) sequencing depth, (**C**) on-target rate, and (**D**) GC content of consensus reads were compared. (Twist = Twist Pan-cancer reference standard, SeraCare = SeraSeq ctDNA complete reference material, Horizon = OncoSpan cfDNA, SeraCare_Myo = SeraSeq Myeloid ctDNA mix, SensID = SensID cfDNA reference material, Coriell = NA24385 gDNA provided by Coriell, patient = patient samples, *p*-value < 0.05: *, *p*-value < 0.01: **).

**Figure 3 cancers-15-05024-f003:**
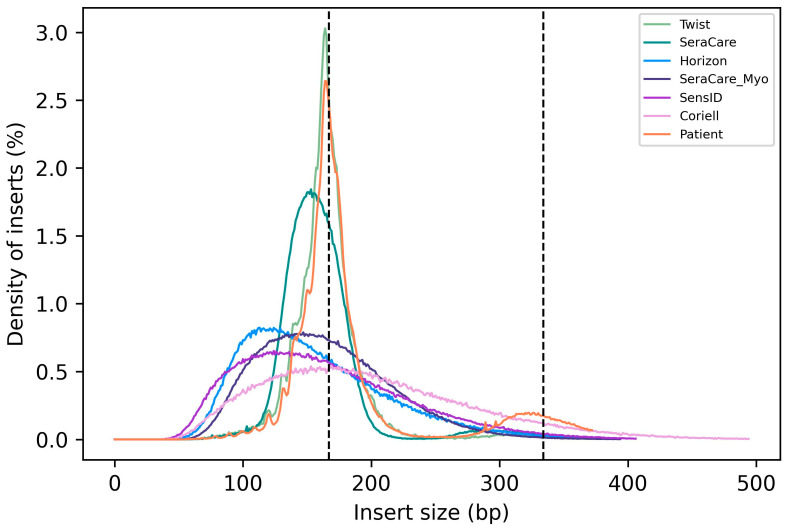
Insert size distribution of cfDNA reference materials and native cfDNA. (Twist = Twist Pan-cancer reference standard, SeraCare = SeraSeq ctDNA complete reference material, Horizon = OncoSpan cfDNA, SeraCare_Myo = SeraSeq Myeloid ctDNA mix, SensID = SensID cfDNA reference material, Coriell = NA24385 gDNA provided by Coriell, patient = patient samples).

**Figure 4 cancers-15-05024-f004:**
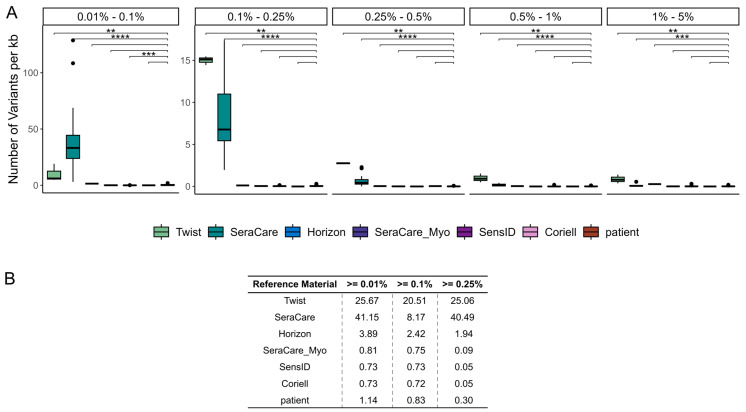
Number of variants per kb with low VAFs in reference materials and patient samples. (**A**) The number of variants identified per kb within distinct VAF intervals and the (**B**) total number of variants identified per kb above the cutoffs of 0.01%, 0.1%, and 0.25% VAF were assessed. (Twist = Twist Pan-cancer reference standard, SeraCare = SeraSeq ctDNA complete reference material, Horizon = OncoSpan cfDNA, SeraCare_Myo = SeraSeq Myeloid ctDNA mix, SensID = SensID cfDNA reference material, Coriell = NA24385 gDNA provided by Coriell, patient = patient samples, *p*-value < 0.01: **, *p*-value < 0.001: ***, *p*-value < 0.0001: ****).

**Figure 5 cancers-15-05024-f005:**
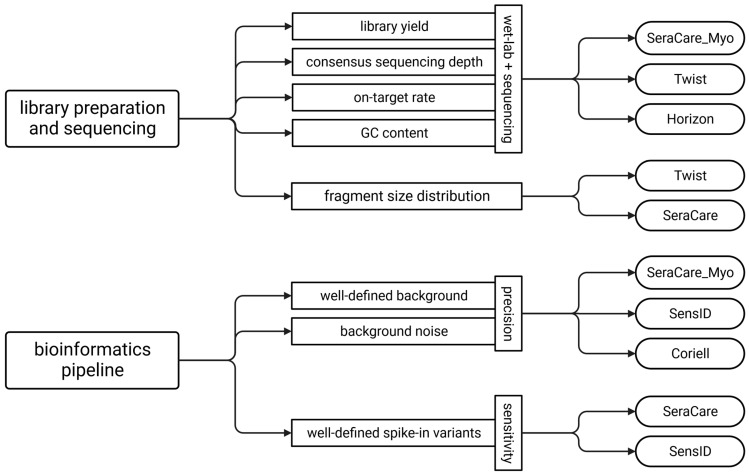
Guideline for the selection of reference materials for analytical performance evaluation of liquid biopsy NGS assays. (Twist = Twist Pan-cancer reference standard, SeraCare = SeraSeq ctDNA complete reference material, Horizon = OncoSpan cfDNA, SeraCare_Myo = SeraSeq Myeloid ctDNA mix, SensID = SensID cfDNA reference material, Coriell = NA24385 gDNA provided by Coriell).

## Data Availability

The data presented in this study are available within the article and its Appendix A files.

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
