# Peer review of "Impact of cfDNA Reference Materials on Clinical Performance of Liquid Biopsy NGS Assays"

_cancers, 2023, doi:10.3390/cancers15205024_

Round 1

Reviewer 1 Report

authors do not consider how VAF in the tested samples might be biased by:

- high range of cfDNA input 5-100 used for library preparation

- presence of gDNA in the cfDNA mixture. Have authors tried to remove the genomic DNA and use it for the analysis size selected cfDNA?

Both aspects impacted on the final output.  Preparing the standardization authors should pay more attention to those contexts that might increase the robustness of the study.

There is no info on what type of variants can be detected in tested reference materials

How authors explain that increasing input of reference material was associated with decreased VAF – figure 4B

The authors did not perform any cross-lab checks on patient samples.

The conclusion should be implemented in future perspectives regarding liquid biopsy testing.

Author Response

Dear Reviewer,

thank you for the revision of our manuscript “Impact of cfDNA reference materials on clinical performance of liquid biopsy NGS assays” We appreciate your time and effort for providing valuable feedback on our initial manuscript.

We considered your suggestions carefully and described the differences in input material and its potential impact on results in more detail. We strongly believe that adapting the manuscript according to your suggestions significantly improved our manuscript.

We respond to all your comments and document the resulting manuscript changes at the bottom of this letter.

Sincerely,

Ariane Hallermayr, PhD

General Comments

Yes

Can be improved

Must be improved

Not applicable

Does the introduction provide sufficient background and include all relevant references?

(  )

(x)

(  )

(  )

Are all the cited references relevant to the research?

(  )

(x)

(  )

(  )

Is the research design appropriate?

(  )

(  )

(x)

(  )

Are the methods adequately described?

(  )

(  )

(x)

(  )

Are the results clearly presented?

(  )

(  )

(x)

(  )

Are the conclusions supported by the results?

(  )

(  )

(x)

(  )

Comment 1: authors do not consider how VAF in the tested samples might be biased by:

(a) high range of cfDNA input 5-100 used for library preparation and

(b) presence of gDNA in the cfDNA mixture. Have authors tried to remove the genomic DNA and use it for the analysis size selected cfDNA?

Both aspects impacted on the final output. Preparing the standardization authors should pay more attention to those contexts that might increase the robustness of the study.

Response: We thank the reviewer for this comment. Thank you for noticing this. Actually, the range of input material used was smaller (5-50 ng).

(a) To ensure comparability of results and standardization across laboratories, we used 25 ng of input material for all cfDNA reference materials. However, since for patient samples we were able to obtain a broad range of cfDNA, we decided to use all available cfDNA as input material, to provide best possible care. In terms of statistics, it is more likely to detect low frequency variants if a higher amount of input DNA was used. However, since we did not observe differences in the number of low frequency variants detected across patient samples, and patient samples with 50 ng of input material still had very low numbers of low frequency variants, we strongly believe that this does not influence our comparison to reference materials.

(b) To avoid presence of gDNA in the cfDNA mixture, we used cfDNA stabilizing tubes for sample collection (Streck or PAXgene). We further confirmed purity of cfDNA be measuring cfDNA samples on the Fragment Analyzer or TapeStation system. If still some germline DNA might be remaining in our samples, gDNA will be lost throughout the process of library preparation. No fragmentation is included in the library preparation workflow, since cfDNA is already natually fragmented to a length of ~167 bp. Therefore during library preparation long gDNA fragments will be removed by size selection using beads.

Changes: (a) To emphasize differences in the amount of input material, we added additional information to the Methods section (track changes: p.4 l.131-132 / clean: p.4 l.129-130). We further discuss the potential impact of the differences in input material to the limitations of our study (track changes: p.11, l.397-404 / clean: p.11, l.395-402).

(b) We added additional information on samples collection tubes used and on quality checking of cfDNA purity to the Methods section (track changes: p.4 l.109-110, l. 118 / clean: p.4 l.108-109, l.117).

Comment 2: There is no info on what type of variants can be detected in tested reference materials

Response: We thank the reviewer for this comment. As described in the Methods section, detailed information on the reference materials used (incl. catalogue numbers) are described in supplementary Table S1.

Changes: To emphasize the types of spike-in variants present in the different reference materials (SNVs and / or InDels) we added this information to the Methods section (track changes: p.3 l.94-95, l.97, l.99, l.104 / clean: p.3 l.93-94, l.96, l.98, l.103)

Comment 3: How authors explain that increasing input of reference material was associated with decreased VAF – figure 4B

Response: We thank the reviewer for this comment. We did not use increasing input of reference material. For all reference materials we used 25 ng of input DNA,

Changes: To emphasize that the same amount of input material was used for reference materials we explained this in more detail in the Methods section (track changes: p.4 l.131-132 / clean: p.4 l.129-130).

Comment 4: The authors did not perform any cross-lab checks on patient samples.

Response: We agree with the reviewer that this is a limitation of our study. In general, only little amount of cfDNA is available from plasma, therefore cross-lab checks on patient samples are almost impossible. Analyzing the same reference materials across labs showed comparable results (track changes: p.11 l.393-396 / clean: p.11 l.392-395). With this we were able to show comparable performance of our assay across labs.

Changes: To emphasize that patient samples were not analyzed across labs we added this to the discussion on limitations of our study (track changes: p.11 l.394 / clean: p.11 l.393).

Comment 5: The conclusion should be implemented in future perspectives regarding liquid biopsy testing.

Response: Thank you for this comment.

Changes: To emphasize that increasing evidence on the clinical utility of liquid biopsy based diagnosis will eventually lead to implementation into clinical routine practice we added a sentence at the end of the Conclusions section (track changes: p.11, l.415-417 / clean: p.11 l. 413-415).

Reviewer 2 Report

In this original manuscript by Hallermayr and colleagues, the authors conduct a comprehensive analysis comparing six different types of commercial cfDNA reference materials. Their objective is to assess the suitability of these materials for quality control in Liquid Biopsy NGS, using a dataset of 42 samples obtained from tumor patients. The study reveals significant disparities between the artificial reference materials and native cfDNA, leading the authors to propose guidelines for the judicious selection of appropriate reference materials at various stages of the quality control process.

Overall, this reviewer commends the authors for addressing a crucial issue within the field and appreciates the balanced approach taken in presenting their findings. The manuscript aligns well with the objectives of the special issue.

A suggestion for improvement would be to consider a more concise title for the article, which could enhance its readability. Additionally, it might be beneficial to expand the initial mention of the acronym NGS (Next-Generation Sequencing) in the text to provide clarity for readers unfamiliar with the abbreviation.

Author Response

Dear Reviewer,

thank you for the revision of our manuscript “Impact of cfDNA reference materials on clinical performance of liquid biopsy NGS assays” We appreciate your time and effort for providing valuable feedback on our initial manuscript.

We thank you for the positive feedback on our manuscript and considered your suggestions carefully. We strongly believe that adapting the manuscript according to your suggestions significantly improved our manuscript.

We respond to all your comments and document the resulting manuscript changes at the bottom of this letter.

Sincerely,

Ariane Hallermayr, PhD

General Comments

Yes

Can be improved

Must be improved

Not applicable

Does the introduction provide sufficient background and include all relevant references?

(x)

(  )

(  )

(  )

Are all the cited references relevant to the research?

(x)

(  )

(  )

(  )

Is the research design appropriate?

(x)

(  )

(  )

(  )

Are the methods adequately described?

(x)

(  )

(  )

(  )

Are the results clearly presented?

(x)

(  )

(  )

(  )

Are the conclusions supported by the results?

(x)

(  )

(  )

(  )

Comment 1: In this original manuscript by Hallermayr and colleagues, the authors conduct a comprehensive analysis comparing six different types of commercial cfDNA reference materials. Their objective is to assess the suitability of these materials for quality control in Liquid Biopsy NGS, using a dataset of 42 samples obtained from tumor patients. The study reveals significant disparities between the artificial reference materials and native cfDNA, leading the authors to propose guidelines for the judicious selection of appropriate reference materials at various stages of the quality control process.

Overall, this reviewer commends the authors for addressing a crucial issue within the field and appreciates the balanced approach taken in presenting their findings. The manuscript aligns well with the objectives of the special issue.

Response: We thank the reviewer for this positive feedback on our manuscript. We are happy to be able to contribute our work to the community with the hope to improve standardization across laboratories.

Comment 2: A suggestion for improvement would be to consider a more concise title for the article, which could enhance its readability. Additionally, it might be beneficial to expand the initial mention of the acronym NGS (Next-Generation Sequencing) in the text to provide clarity for readers unfamiliar with the abbreviation.

Response: We thank the reviewer for this comment. We agree that a more concise title would enhance the articles readability.

Changes: As suggested we adapted the title and introduced NGS as abbreviation (track changes: p.1 l.2-4, l.19-20, p.2 l.70 / clean: p.1 l.2-3, l.18-19, p.2 l.69)

Reviewer 3 Report

Very important research showing how the choice of reference material can affect the obtained results. The results are presented clearly and thoroughly. Conclusions supported by evidence.

Author Response

Dear Reviewer,

thank you for the revision of our manuscript “Impact of cfDNA reference materials on clinical performance of liquid biopsy NGS assays” We appreciate your time and effort for providing valuable feedback on our initial manuscript.

We thank you for the positive feedback on our manuscript.

We respond to all your comments and document the resulting manuscript changes at the bottom of this letter.

Sincerely,

Ariane Hallermayr, PhD

General Comments

Yes

Can be improved

Must be improved

Not applicable

Does the introduction provide sufficient background and include all relevant references?

(x)

(  )

(  )

(  )

Are all the cited references relevant to the research?

(x)

(  )

(  )

(  )

Is the research design appropriate?

(x)

(  )

(  )

(  )

Are the methods adequately described?

(x)

(  )

(  )

(  )

Are the results clearly presented?

(x)

(  )

(  )

(  )

Are the conclusions supported by the results?

(x)

(  )

(  )

(  )

Comment 1: Very important research showing how the choice of reference material can affect the obtained results. The results are presented clearly and thoroughly. Conclusions supported by evidence.

Response: We thank the reviewer for this positive feedback on our manuscript. We are happy to be able to contribute our work to the community with the hope to improve standardization across laboratories.

Round 2

Reviewer 1 Report

The authors responded to all remarks in a satisfactory way. The paper can be accepted.